# Abstractified Multi-instance Learning (AMIL) for Biomedical Relation Extraction

**William Hogan**[1,2]                                                            WHOGAN@UCSD.EDU
**Molly Huang**[2]                                                               M7HUANG@UCSD.EDU
**Yannis Katsis**[3]                                                         YANNIS.KATSIS@IBM.COM
**Tyler Baldwin**[3]                                                         TBALDWIN@US.IBM.COM
**Ho-Cheol Kim**[3]                                                              HCKIM@US.IBM.COM
**Yoshiki Vazquez Baeza**[2]                                                      YOSHIKI@UCSD.EDU
**Andrew Bartko** [2]                                                            ABARTKO@UCSD.EDU
**Chun-Nan Hsu**[4]                                                              CHUNNAN@UCSD.EDU

[1]*Department of Computer Science & Engineering, University of California, San Diego*

[2]*Center for Microbiome Innovation, University of California, San Diego*

[3]*IBM Research-Almaden*

[4]*Center for Research in Biological Systems, University of California, San Diego*

## Abstract

Relation extraction in the biomedical domain is a challenging task due to a lack of labeled data and a long-tail distribution of fact triples. Many works leverage distant supervision which automatically generates labeled data by pairing a knowledge graph with raw textual data. Distant supervision produces noisy labels and requires additional techniques, such as multi-instance learning (MIL), to denoise the training signal. However, MIL requires multiple instances of data and struggles with very long-tail datasets such as those found in the biomedical domain. In this work, we propose a novel reformulation of MIL for biomedical relation extraction that abstractifies biomedical entities into their corresponding semantic types. By grouping entities by types, we are better able to take advantage of the benefits of MIL and further denoise the training signal. We show this reformulation, which we refer to as abstractified multi-instance learning (AMIL), improves performance in biomedical relationship extraction. We also propose a novel relationship embedding architecture that further improves model performance.

## 1. Introduction

Relation extraction (RE) is a key facet of information extraction in large bodies of unstructured textual data. RE is particularly important in the biomedical domain where extracting relationships between pairs of biomedical entities, also known as "fact triples", can produce new insights into complicated biological interactions. For instance, with the near-exponential growth of microbiome research [Sa'ed et al., 2019], advanced RE methods may help discover important links between gut microbiota and diseases. It is in this context that we motivate our work.

RE within the biomedical domain comes with two inherent challenges: there are more than 30 million scientific articles, with hundreds of thousands of articles published every year, and there is a corresponding lack of labeled data. To resolve these challenges, many have leveraged distant supervision techniques which pair knowledge graphs with raw textual data to automatically generate labels to train deep-learning models [Gu et al., 2019, Su et al., 2019, Junge and Jensen, 2019]. We seek to improve distantly supervised biomedical RE methods in this work. We use the Unified Medical Language System (UMLS) Metathesaurus [Bodenreider, 2004] for our knowledge graph and pair it with raw textual data from PubMed [Canese and Weis, 2013].

To automatically generate labels, distantly supervised RE methods rely on a simple yet powerful assumption: any singular sentence that contains a pair of entities also expresses a relationship, as determined by the accompanying knowledge graph, between those entities [Mintz et al., 2009].

However, this assumption leads to a noisy training signal with many false positives as not all sentences express a relationship between an entity pair. To combat this, many works have leveraged multi-instance learning (MIL) [Riedel et al., 2010, Hoffmann et al., 2011, Zeng et al., 2015] where, instead of assessing single sentences, MIL assesses positive and negative *bags* of sentences that contain the same entity pair. Grouping sentences into bags greatly reduces noise in the training signal since a bag of sentences is more likely to express a relationship than a single sentence. This enables the model to better classify relationships between unseen entity pairs.

However, similar to many NLP tasks, biomedical RE suffers from a long-tail distribution of fact triples, where many entity pairs are only supported by a few sentences of evidence. After processing the PubMed corpus, we observe that a majority ($\sim 52\%$) of extracted triples are supported by fewer than three sentences. Creating bags of sentences for such entity pairs requires heavy up-sampling. For example, if a pair of entities is only supported by one sentence and a bag size is equal to 16 sentences, the single sentence is duplicated 15 times to fill the bag. This erases the benefit of MIL. To counter this issue, we introduce abstractified multi-instance learning (AMIL) where, instead of grouping entity pairs by name, we group entities by the corresponding semantic type as determined by UMLS. UMLS categorizes each entity with a semantic type within the UMLS semantic network. The UMLS semantic network is curated by human experts for decades and provides a rich ontology of biomedical concepts which we leverage to group multiple different entity pairs within a single MIL bag, reducing the need to up-sample sentences.

For example, consider two sentences: (1) a sentence containing the entity pair (*fibula*, *tibia*) and (2) a second sentence containing the entity pair (*humerus*, *ulna*). With distant supervision, we assume each sentence expresses the relationship linking both pairs, namely *articulates with*. Despite expressing the same relationship, without abstraction, these sentences are placed into separate MIL bags since bags are grouped by distinct entity pairs. By introducing abstractified multi-instance learning, the entities *fibula*, *tibia*, *humerus*, and *ulna* are grouped by their corresponding UMLS semantic type—"*Body Part, Organ, or Organ Component.*" This allows us to place the aforementioned sentences into the same MIL bag based on their entity type, creating a heterogeneous bag of entity pairs that express the same relationship.

With this reformulation, bags containing a single duplicated sentence are reduced by half. AMIL produces better overall performance for biomedical RE with significant performance gains for "rare" triples. Here, we define "rare" triples as triples that are supported by fewer than eight sentences. These triples make up roughly 80% of the long-tail distribution of triples.

We also take inspiration from Soares et al.(2019) and conduct a suite of experiments with variations of relationship embedding architectures. Such experiments are underexplored in the biomedical domain and many are novel to the general task of relationship classification. Soares et al. report the best RE performance using a relationship representation consisting of embedded entity start markers—special span tokens that denote the beginning of an entity. We test this RE architecture in the biomedical domain and also test the performance of entity end markers. Moreover, we introduce a novel relationship representation, namely the middle mention pool, which pools word pieces between head and tail entities. This embedding architecture is inspired by the observation that context between two biomedical entities in a sentence often contains the information-rich and relationship-relevant signal.

Our best performing relationship embedding architecture results from the combination of both entity end markers and the middle mention pool. We observe that this architecture further increases the performance of our relation classification model.

In this paper, we make the following contributions:

- We introduce abstractified multiple-instance learning (AMIL), which achieves new state-of-the-art performance for biomedical relationship extraction. We also report significant performance gains for rare fact triples.

- We propose an improved relationship representation for biomedical relation extraction. We show that concatenating embedding tokens from entity end markers with the middle mention pool produces the best performing model.

- We make all our code, saved models, and pre-processing scripts publicly available[1] to facilitate future biomedical RE efforts. Pre-processing scripts can impact model performance and are important to prepare an up-to-date, ready-for-RE dataset from ever-growing PubMed and UMLS. Our results in Section 5 show that using updated pre-processing tools can improve model performance by $\sim 10\%$.

## 2. Related Work

Early works combining distant supervision with relation extraction [Bunescu and Mooney, 2007, Craven and Kumlien, 1999] relied on the strong assumption claiming that, if a relationship between two entities exists, then all sentences containing those entities express the corresponding relationship. This assumption was relaxed by Riedel et al. (2010) with the introduction of multi-instance learning (MIL) which claimed that if a relationship exists between two entities, at least one sentence that contains the two entities may express the corresponding relation. Hoffmann et al. (2011) build on the work of Riedel et al. by allowing for overlapping relations. Zeng et al. (2015) extend distantly supervised RE by combining MIL with a novel piecewise convolutional neural network (PCNN).

Lin et al. (2016) made further improvements by introducing an attention mechanism that attends to relevant information in a bag of sentences. This sentence-level attention mechanism for MIL inspired numerous subsequent works [Luo et al., 2017, Han et al., 2018a, Alt et al., 2019]. Han et al. (2018a) propose a joint training RE model that combines a knowledge graph with an attention mechanism and MIL. Dai et al. (2019) extend the work by Han et al. into the biomedical domain and use a PCNN for sentence encoding. Amin et al.(2020) propose an RE model that uses BioBERT [Lee et al., 2019], a pretrained transformer based on BERT [Devlin et al., 2019], for sentence encoding. They leverage MIL with entity-marking methods following R-BERT [Wu and He, 2019] and achieve the best performance when the directionality of extracted triples is matched to the directionality from the UMLS knowledge graph. Notably, the model proposed by Amin et al.(2020) does not benefit from sentence-level attention. We choose the model proposed by Amin et al.(2020) for our baseline model as it achieves the current state-of-the-art (SOTA) in biomedical RE.

We also conduct a suite of experiments with variations of relationship embedding architectures. These experiments are inspired by Soares et al. (2019) who conduct experiments with six different embedding architectures and report performance on general-domain RE datasets. They show that constructing a relationship embedding with special entity start markers outperforms other architectures. We build on this work by (1) conducting similar experiments in the biomedical domain and (2) by proposing numerous novel architectures for a total of seventeen alternatives for a comprehensive comparison on the biomedical RE task.

## 3. Datasets

**UMLS Metathesaurus and Semantic Network**: The UMLS Metathesaurus and Semantic Network is a knowledge graph of biomedical entities and their corresponding relationships. Following numerous previous works in biomedical relation extraction [Zhang and Wang, 2015, Dai et al., 2019, Amin et al., 2020], we only extract fact triples that contain a relationship other than "synonymous", "narrower", or "broader". These general relationships make up the majority of relationships in the UMLS Semantic Network and we exclude them to focus on more substantive relationships. Using this filter, we extract 7,025,733 triples from the 2019AB UMLS release.

---

1. https://github.com/IBM/aihn-ucsd/tree/master/amil

**PubMed**: For our textual data, we use abstracts from the 2019 PubMed corpus[2]. The corpus contains 34.4M abstracts which we segment into 158,848,048 unique sentences using Sci-Spacy [Neumann et al., 2019].

## 4. Method

**Problem statement:** given knowledge graph $\mathcal{G}$ and text corpus $\mathcal{C}$, sentences $s$ from $\mathcal{C}$ that contain exactly two distinct entities, $(e^i_1, e^i_2)$, that are linked via relationship $r^i$ as determined by $\mathcal{G}$, are grouped into bags $B^i = \{s^i_1, \ldots, s^i_m\}$ based on their corresponding entity types $(e^i_{1-Type}, e^i_{2-Type})$ where $e^i_{1-Type}, e^i_{2-Type} \in \mathcal{E}_{\mathcal{T}}$ and $\mathcal{E}_{\mathcal{T}}$ represents the set of all UMLS semantic types. Our goal is to predict the relationship $r^i$ that is expressed in each bag $B^i$, forming a fact triple $\mathcal{T} = \{(e^i_1, r^i, e^i_2)\}$. For simplicity, indices for bags, sentences, and entities are omitted if not required for clarity.

### 4.1 Pre-processing

Although static benchmark test sets are typically available for general-domain RE tasks (e.g., [Zhang et al., 2017, Han et al., 2018b]), there are no such test sets for biomedical RE. Biomedical RE relies on large datasets that are constantly updated (e.g. PubMed and UMLS) and, part of the task involves developing a pre-processing pipeline. To best compare the performance of our approach, we model our pre-processing steps after those used by Amin et al. [2020]. Entities within sentences are found using the UMLS Metathesaurus which contains every UMLS concept and their corresponding surface form variations. Sentences are retained and considered "positive" examples if they meet the following criteria: (1) they contain exactly two distinct entities and (2) those entities are linked by a relationship in the UMLS knowledge graph.

Each sentence is grouped by its corresponding fact triple $\mathcal{T} = \{(e_1, r, e_2)\}$, where $e_1, e_2 \in \mathcal{E}$ and $\mathcal{E}$ represents the set of UMLS entities and $r \in \mathcal{R}$ where $\mathcal{R}$ is the set of UMLS relation types. The UMLS knowledge graph provides directionality for relationships (i.e. directed edges) and that directionality preserved in the fact triples extracted from sentences regardless of the order entities appear in a sentence. Amin et al. [2020] show that preserving directionality further denoises the training signal and leads to better predictive performance.

Negative examples are generated by randomly replacing either a head or tail entity within a positive sentence such that the newly formed entity pair is not linked by a relationship in the UMLS knowledge graph. The entity pairs extracted from negative sentences are assigned the negative relationship label "NA" to form negative triples. Triples from the negative class are chosen randomly and the size of the negative class is set to 70% of the largest positive relationship class.

Span markers denoting the start and end of entity spans are then inserted into each sentence. Head entities $(e_1)$ are marked with '^' and tail entities $(e_2)$ are marked with '$'. Soares et al. show that BERT achieves the best sentence-level relationship extraction performance when entity spans are denoted with special start and end tokens.

We construct random train/dev/test splits based on extracted triples and ensure no triples or sentences overlap between the splits. We use 20% of the data for a test set. With the remaining 80% of data, 10% is used for a development set and the rest is used for training. These steps mirror those used by Amin et al.(2020) but our splits contain different sets of triples and sentences. To ensure fair comparison, we trained the Amin et al.(2020) model, AMIL, and all AMIL variations on identical data from our randomized splits.

Lastly, to train AMIL, entities are abstracted using their corresponding entity-types (immediate hypernyms), which are determined using the UMLS Semantic Network, and grouped into more general entity-type triples, $\{(e_{1-Type}, r, e_{2-Type})\}$. Bags of sentences are then formed using the abstracted entity types and sentences are randomly up-sampled to fill any bags that fall short of the set bag size of 16 sentences. An example of an abstractified bag is provided in Figure 1(b).

---

2. https://mbr.nlm.nih.gov/Download/Baselines/2019/

|       | Num. Sentences | Num. Triples |
|-------|----------------|--------------|
| Train | 647,408        | 64,817       |
| Dev   | 134,768        | 8,423        |
| Test  | 326,128        | 20,383       |

Table 1: Total number of positive and negative example sentences and triples in each split.

### 4.2 Training

We use a pretrained transformer, namely BioBERT [Lee et al., 2019], to produce low-dimensional relationship embeddings with the following hyper-parameters:

- **Transformer Architecture**: 12 layers, 12 attention heads, 786 hidden size
- **Weight Initialization**: BioBERT
- **Activation**: GELU [Hendrycks and Gimpel, 2016]
- **Learning Rate**: 2e-5 with Adam
- **Batch Size**: 2
- **Max sequence length:** 128
- **Total Parameters:** 110M

Each bag of sentences containing entity markers is passed through the transformer to obtain an encoded sentence. For our baseline AMIL model, we match our relationship representation architecture to that used by Amin et al. (2020). We first condense each encoded head and tail entity via average pooling, where $(j, k)$ is the span containing the head entity $e1$, $(l, m)$ is the span containing the tail entity $e2$, and an encoded sentence of length $n$ is denoted as $[(h_0, ..., h_n)]$:

$$\mathbf{h_{e1}} = \frac{1}{1 + k - j} \sum_{i=j}^{k} \mathbf{h}_i$$

$$\mathbf{h_{e2}} = \frac{1}{1 + m - l} \sum_{i=l}^{m} \mathbf{h}_i$$

Pooled entities are then concatenated with the `[CLS]` embedding to form the relationship representations for each sentence in the bag $\mathbf{r} = \langle h_{CLS} | h_{e1} | h_{e2} \rangle \in \mathbb{R}^{3d}$, where $\langle x | y \rangle$ denotes the concatenation of $x$ and $y$. The representations are then aggregated via average pooling and sent through a tanh activation, a dropout layer, and, finally, a fully connected $(2304 \times 2304)$ linear layer. We use cross-entropy to compute the loss and train the model over 300 epochs with early-stopping on the best F1-score from the development set.

All models were trained on an NVIDIA Tesla V100 and completed with an average training time of 7 hours 59 minutes.

### 4.3 Relationship Representations

We present a suite of experiments with various relationship embedding architectures. We draw inspiration from Soares et al. (2019) and conduct 17 experiments to empirically determine the most effective relationship representation architecture for the task of biomedical relationship classification. We expand on their experiments with 12 novel relationship embedding architectures (types 'F' through 'Q') which we describe in the following section.

The following relation embedding experiments are grouped into pairs that feature the same relationship embedding with and without the reserved `[CLS]` token from BERT. In all experiments,

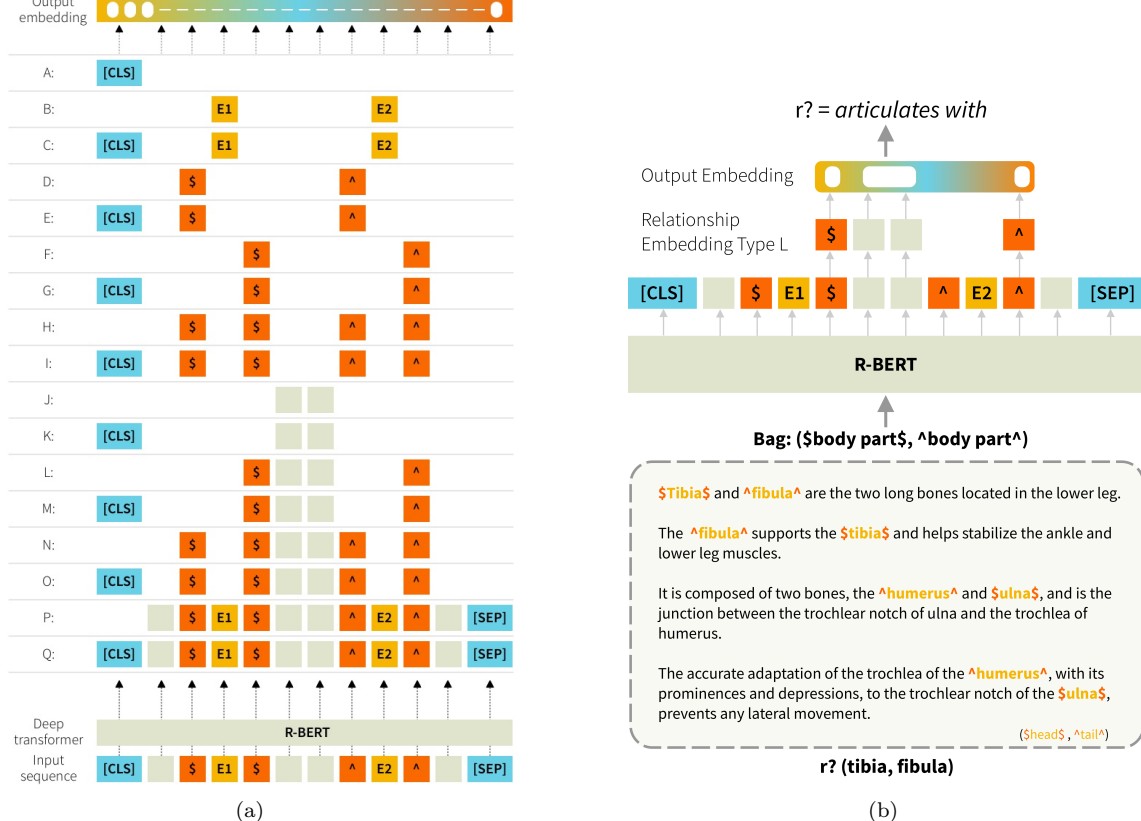

Figure 1: (a) Relation embedding architectures A-Q. Each architecture is defined in Section 4.3.1. (b) Example of data flow in AMIL using relationship type "L" and a bag of sentences grouped by entity type, namely *body part*.

the models are trained using the hyper-parameters and methods described in Section 4.2 (hidden-dimension $d = 786$). All experiments that involve pooling are conducted using average pooling. Figure 1(a) provides a visual representation of each relation embedding architecture and Figure 1(b) illustrates the flow of data using AMIL and an abstractified bag of sentences.

### 4.3.1 Types of Relationship Representations

**A — [CLS] Token**: The [CLS] token from BERT acts as a representation of the entire input sequence and, thus, serves as a baseline for our experiments with various relationship representation architectures.

$$\mathbf{r}_A = \langle h_{CLS} \rangle \in \mathbb{R}^d$$

**B, C — Entity Mention Pool**: For this architecture, each entity's word pieces are pooled via average pooling and then concatenated together. Architecture type 'C', which contains the entity mention pool concatenated [CLS] token, is the architecture used by both our baseline AMIL model and the model proposed by Amin et al.(2020) It achieves the current SOTA in biomedical relation

classification.

$$\mathbf{r}_B = \langle h_{e1} | h_{e2} \rangle \in \mathbb{R}^{2d}, \quad \mathbf{r}_C = \langle h_{CLS} | h_{e1} | h_{e2} \rangle \in \mathbb{R}^{3d}$$

**D, E — Entity Start Markers**: Soares et al. report the best relation classification performance using concatenated entity start markers. We recreate this top-performing architecture to test its performance in the biomedical domain.

$$\mathbf{r}_D = \langle h_{e1_S} | h_{e2_S} \rangle \in \mathbb{R}^{2d}, \quad \mathbf{r}_E = \langle h_{CLS} | h_{e1_S} | h_{e2_S} \rangle \in \mathbb{R}^{3d}$$

**F, G — Entity End Markers**: We propose representing the relationship by using the entity end markers to determine if there is any benefit over the entity start markers.

$$\mathbf{r}_F = \langle h_{e1_E} | h_{e2_E} \rangle \in \mathbb{R}^{2d}, \quad \mathbf{r}_G = \langle h_{CLS} | h_{e1_E} | h_{e2_E} \rangle \in \mathbb{R}^{3d}$$

**H, I — Entity Start and End Marker**: For this experiment, we concatenate the representations for both the entity start markers and the entity end markers.

$$\mathbf{r}_H = \langle h_{e1_S} | h_{e1_E} | h_{e2_S} | h_{e2_E} \rangle \in \mathbb{R}^{4d}, \quad \mathbf{r}_I = \langle h_{CLS} | h_{e1_S} | h_{e1_E} | h_{e2_S} | h_{e2_E} \rangle \in \mathbb{R}^{5d}$$

**J, K — Middle Mention Pool**: Here, we propose using the middle mention pool which is the pooled word pieces between the head and tail entities. This embedding architecture is inspired by a pattern we observe in relationship-containing sentences where, often, the context between two entities in a sentence contains the most information-rich and relationship-relevant signal.

$$\mathbf{r}_J = \langle h_M \rangle \in \mathbb{R}^{d}, \quad \mathbf{r}_K = \langle h_{CLS} | h_M \rangle \in \mathbb{R}^{2d}$$

**L, M — Middle Mention Pool and Entity End Markers**: This architecture concatenates the middle mention pool with the entity end markers.

$$\mathbf{r}_L = \langle h_{e1_E} | h_M | h_{e2_E} \rangle \in \mathbb{R}^{3d} \quad (1), \quad \mathbf{r}_M = \langle h_{CLS} | h_{e1_E} | h_M | h_{e2_E} \rangle \in \mathbb{R}^{4d}$$

**N, O — Middle Mention Pool and Entity Start and End Markers**: We form this representation by concatenating entity start markers, entity end markers, and the middle mention pool. This architecture results in the highest dimensional relationship representation and will help us determine if the added information is beneficial to model performance.

$$\mathbf{r}_N = \langle h_{e1_S} | h_{e1_E} | h_M | h_{e1_S} | h_{e2_E} \rangle \in \mathbb{R}^{5d}, \quad \mathbf{r}_O = \langle h_{CLS} | h_{e1_S} | h_{e1_E} | h_M | h_{e1_S} | h_{e2_E} \rangle \in \mathbb{R}^{6d}$$

**P, Q — Complete Sequence Pool**: We obtain the average of all the output tokens to form the complete sequence pool. This is different from the `[CLS]` token, which is a learned representation of a sequence, in that it averages all the encoded word piece tokens in a sequence. All other relationship representations consist of subsets of the output tokens. By including all tokens, this representation acts as a type of baseline experiment that will allow us to validate or invalidate the use of subsets in other architectures.

$$\mathbf{r}_P = \langle h_{Seq.Avg.} \rangle \in \mathbb{R}^{d}, \quad \mathbf{r}_Q = \langle h_{CLS} | h_{Seq.Avg.} \rangle \in \mathbb{R}^{2d}$$

### 4.4 Evaluation

All evaluations between AMIL and Amin et al.(2020) are conducted using identical sets of triples and sentences. To properly evaluate AMIL, we first de-abstract the triples in a bag of sentences and evaluate performance on the original set of triples. Our pre-processing steps vary from Amin et al.(2020) in that we segment sentences using Sci-Spacy [Neumann et al., 2019] instead of NLTK [Loper and Bird, 2002]. Sci-Spacy is specifically tuned to process biomedical texts where as NLTK is tuned for general English. Using Sci-Spacy, we observe a  30% reduction in extracted sentences due to

fewer extracted sentence fragments. We train the Amin et al.(2020) model using our improved pre-processing steps which results in a higher performance than reported in their original paper ($\sim 10\%$ increase in AUC).

We were unable to attain the code and data used by Dai et al.(2019). Ideally, we would have trained and tested the Dai et al.(2019) model with the same data we used for our other experiments. Since this was not an option, we provide the results of the Dai et al.(2019) model as reported by the authors. Without access to data from their experiments, we believe a direct comparison is not fair; however, the precision @$k$ indicates the model's overall ability to extract true triples from a hold-out set of triples found in a test corpus. Because we use similar data (e.g. the UMLS knowledge graph with raw text from PubMed abstracts), we believe this metric allows for a good, but not perfect, comparison.

We evaluate our model using both corpus-level and sentence-level evaluation:

**Corpus-level Evaluation**: The benchmarks for biomedical RE are set using a corpus-level evaluation [Mintz et al., 2009] which evaluates model performance on a hold-out set of triples. Using this method, we sort predictions of triples from the test set based on their softmax probability and compare the set of predicted triples to the ground truth triples contained in the test corpus. We then report AUC, F1, and precision on the top $K$ predictions.

**Sentence-level Evaluation**: To better understand model performance, we decompose triples into two groups: (1) "rare" triples and (2) "common" triples and conduct a sentence-level evaluation. Extracted triples follow a heavy-tailed Pareto distribution. Using the Pareto principle, we define rare triples as triples that make up the lower 80% of the long-tail distribution of extracted triples. These are triples that are supported by seven or fewer sentences. Common triples are defined as triples that make up the upper 20% of the long-tail distribution of triples and are supported by eight or more sentences. Sentence-level evaluation relies on standard precision, recall, and F1-score metrics. It allows us to more holistically assess model performance since sentence-level metrics do not obscure performance on low-confidence predictions. A relationship predicted between an entity pair that matches the ground truth relationship, as determined by the UMLS knowledge graph, is a true positive. A relationship predicted between an entity pair that is not linked in the UMLS knowledge graph is a false positive. A false negative occurs when the model predicts "NA" and the ground truth is something other than "NA." Note, sentence-level evaluation is not a good estimate of overall model performance since, for this task, we are primarily interested in knowledge-graph completion. We only include sentence-level evaluation to evaluate AMIL's ability to predict rare triples. Corpus-level evaluation is more appropriate for assessing overall model performance.

## 5. Results

Table 2 compares the performance of our baseline AMIL model to other distantly supervised biomedical relation extraction models. We also include performance from our top-performing relationship embedding architecture—relationship representation architecture type 'L'. Here, we observe AMIL achieves significantly higher scores for AUC and F1. It also outperforms on each subset of precision. AMIL using relationship representation architecture type 'L' makes additional gains in each metric.

Table 3 compares sentence-level performance on the rare and common subsets of triples. We also include sentence-level performance on all triples for comparative purposes. Again, AMIL outperforms Amin et al.(2020) in all metrics. AMIL with relation representation type 'L' attains an additional performance boost in all metrics but recall for common triples.

Table 4 compares the 17 variations of relationship embedding architectures described in Section 4.3 using a corpus-level evaluation. We observe that relationship embedding architecture type 'L' outperforms other architectures in all metrics.

| RE Model | AUC | F1 | P@2k | P@4k | P@6k | P@10k | P@20k |
|---|---|---|---|---|---|---|---|
| Dai et al. (2019)* | — | — | .913 | .829 | .753 | — | — |
| Amin et al. (2020)* | .684 | .649 | .983 | .977 | .969 | — | — |
| Amin et al. (2020) w/Sci-Spacy | .758 | .710 | .998 | .995 | .991 | .981 | .905 |
| AMIL | .862 | .795 | .997 | .997 | .997 | .994 | .947 |
| AMIL Rel. Type 'L' | **.872** | **.812** | **1.000** | **.999** | **.999** | **.995** | **.953** |

Table 2: Corpus-level performance of AMIL versus the other distantly supervised biomedical relation extraction models. '*' denotes performance as reported by the original authors otherwise the results are from our own implementation. Note, as explained in Section 4.4, due to a difference in data, models with '*' are not directly comparable to other models. AMIL with relationship type 'L' is defined by Equation (1) in Section 4.3.1.

| RE Model | P | R | F1 |
|---|---|---|---|
| **All Triples** | | | |
| Amin et al. (2020) | .635 | .634 | .635 |
| AMIL | .728 | .727 | .727 |
| AMIL Rel. Type 'L' | **.740** | **.733** | **.737** |
| **Rare Triples** | | | |
| Amin et al. (2020) | .625 | .624 | .624 |
| AMIL | .729 | .729 | .729 |
| AMIL Rel. Type 'L' | **.746** | **.738** | **.742** |
| **Common Triples** | | | |
| Amin et al. (2020) | .679 | .677 | .678 |
| AMIL | .724 | **.720** | .722 |
| AMIL Rel. Type 'L' | **.726** | .719 | **.723** |

Table 3: Sentence-level performance on rare triples (lower 80% of the long-tail distribution of triples) and common triples (upper 20% of the long-tail distribution). Rare triples are supported by seven or fewer sentences and common triples are supported by eight or more sentences. AMIL with relationship type 'L' is defined by Equation (1) in Section 4.3.1.

## 6. Discussion

The large performance gains reported in Table 2 confirm that abstractified multi-instance learning is successful in further denoising the training signal for distantly supervised relation extraction. By grouping entities by entity type, we are able to better leverage the benefits of multi-instance learning on long-tail datasets.

We hypothesize that AMIL as a denoising strategy will have the greatest impact on rare triples. Rare triples and their corresponding sentences require the most up-sampling to fill a bag of sentences and, thus, should receive a greater benefit compared to common triples when grouped into larger entity-type bags. The results in Table 3 confirm this hypothesis. Compared to Amin et al.(2020) which uses standard MIL, AMIL gains 10.5 F1 percentage points on rare triples compared to a 4.4 point gain on common triples.

From table 4, the high performance of relationship representation types 'L' and 'J', both of which contain the middle mention pool, confirms our hypothesis that the context between two entities

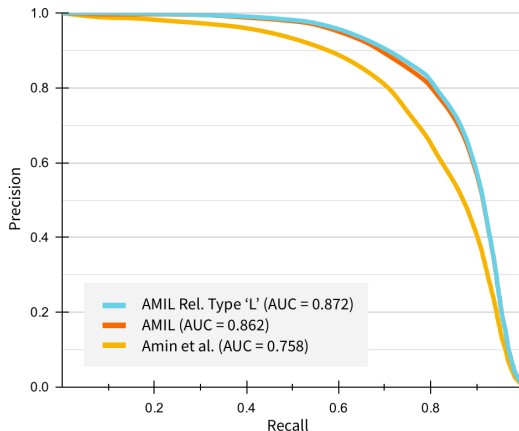

Figure 2: Aggregate precision/recall curves of RE models. AMIL makes large gains over Amin et al.(2020), and AMIL using relationship representation type 'L' makes additional gains.

| | Relationship Representation | F1 | AUC | P@20k |
|---|---|---|---|---|
| A: | `[CLS]` | .793 | .863 | .947 |
| B: | entity mention pool | .786 | .855 | .943 |
| C: | `[CLS]` + entity mention pool | .795 | .862 | .947 |
| D: | $e_{1Start} + e_{2Start}$ | .795 | .859 | .948 |
| E: | `[CLS]` + $e_{1Start} + e_{2Start}$ | .792 | .860 | .946 |
| F: | $e_{1End} + e_{2End}$ | .804 | .872 | .951 |
| G: | `[CLS]` + $e_{1End} + e_{2End}$ | .799 | .861 | .950 |
| H: | $e_{1Start} + e_{1End} + e_{2Start} + e_{2End}$ | .792 | .857 | .947 |
| I: | `[CLS]` + $e_{1Start} + e_{1End} + e_{2Start} + e_{2End}$ | .780 | .859 | .949 |
| J: | middle mention p. | .805 | .862 | .952 |
| K: | `[CLS]` + middle mention p. | .788 | .850 | .945 |
| L: | $e_{1End}$ + middle mention pool + $e_{2End}$ | **.812** | **.872** | **.953** |
| M: | `[CLS]` + $e_{1End}$ + middle mention p. + $e_{2End}$ | .804 | .865 | .951 |
| N: | $e_{1Start} + e_{1End}$ + middle mention p. + $e_{1End} + e_{2End}$ | .800 | .865 | .950 |
| O: | `[CLS]` + $e_{1Start} + e_{1End}$ + middle mention p. + $e_{1End} + e_{2End}$ | .804 | .865 | .950 |
| P: | entire sequence avg | .800 | .862 | .948 |
| Q: | `[CLS]` + entire sequence avg | .808 | .864 | .949 |

Table 4: A comparison of relation embedding architectures used to classify biomedical relationships.

in a sentence provides an information-rich and relationship-relevant signal. Soares et al. report relationship representation type 'D', the entity start markers, as their top-performing architecture as tested on general-domain data. However, our tests in the biomedical domain show that this architecture fails to reach high performance compared to other architectures. This points to the potential need for domain-specific relationship architectures. Comparing the performance of entity *start* markers 'D' with entity *end* markers 'F', we see that the model benefits from the encoding of the end entity markers indicating that, although BERT is bidirectional, the position of embedded special markers informs the model's ability to classify relationships.

We constructed pairs of experiments with and without the special [CLS] token from BERT to determine the effect of the [CLS] token on the model's ability to predict relationships. Interestingly, we observe mixed effects. There is an even split between the performance of architectures with and without a concatenated [CLS] token. Experiment pairs (B, C), (H, I), (L, M), and (P, Q) benefit from the [CLS] token while experiment pairs (D, E), (F, G), (J, K), and (N,O) are hindered by the [CLS] token. In the experiment pairs where the [CLS] token was beneficial, the average increase on the AUC is 0.0075. On experiment pairs that resulted in hindered performance, the average decrease on the AUC is $-0.003$.

Lastly, the performance of the entire sequence average (representations P and Q) justifies the need for subset representations. The model performs best when an information-rich subset of tokens, such as the middle mention pool and/or the entity end markers, are used to construct a relationship representation.

## 7. Conclusion

In this work, we propose abstractified multi-instance (AMIL), a novel denoising method that increases the efficacy of multi-instance learning in the biomedical domain. With it, we improve performance on biomedical relationship extraction and report significant performance gains on rare fact triples. We also propose a novel relationship embedding architecture which further increases model performance.

For future work, we will explore combining AMIL with more advanced bag aggregation methods. We will also explore applying our novel relationship embedding architectures to relationship extraction tasks using general-domain datasets.

## Acknowledgments

Thank you to the anonymous reviewers for their thoughtful comments and corrections. This work is supported by IBM Research AI through the AI Horizons Network.

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
