# OpenReview forum: "Abstractified Multi-instance Learning (AMIL) for Biomedical Relation Extraction"
_AKBC.ws/2021/Conference — AKBC 2021_

### Official Review · Reviewer_Jwtr · 2021-07-20
**Interesting contribution on distantly supervised relation extraction for biomedical domain**

**Rating:** 6
**Confidence:** 4

**Review:**

Summary
This papers studies distantly supervised biomedical RE, where the data have a long-tail distribution.
The authors hypothesize that formulating bags using distinct entity pairs results in many bags with too few sentences, which may hinder performance.

The authors then propose abstractified multi-instance learning (AMIL) that formulates bags using UMLS semantic type (i.e., biomedical named entity type). AMIL outperforms the previous state-of-the-art method in Amin et al. 2020.  Most of the performance gain comes from "rare" tuples, validating the hypothesis in the beginning. The author also empirically compared 17 different methods to construct the representation, where they find the type L achieves the best performance.

Strength
- The proposed method AMIL is a intuitive and effective extension to MIL targeting long-tail data distribution in distantly supervised RE. AMIL outperforms previous state-of-the-art in Biomedical RE.
- The authors did a careful analysis on pre-processing and construction of representation. These small decisions actually influence performance a lot. Such empirical advice can be useful for other researchers.
- The paper is well-organized. The figures are clear and informative.

Weakness
- Dataset creation and evaluation protocol may have defects. See the question below.
- The work may be further strengthened with additional baselines. Since AMIL benefits from the UMLS semantic types, one baseline method I can think of is to include trainable embeddings for these types and include them in constructing relation representation (which is used in pre-BERT methods such as PCNN).

Question
- Since the construction of bag is different in AMIL, is it possible that sentence-level attention (Lin et al., 2016) becomes beneficial for AMIL?
- I wonder whether sentence-level evaluation is reasonable and whether it gives a good estimate of the performance. The test set is taken from a bigger dataset constructed with distant supervision, so the test data can still be very noisy.
- I'm not sure if listing performance from Dai et al. (2019) and Amin et al. (2020) is reasonable. The dataset construction in this paper seems to be different, so the numbers are not comparable.

---

> ### Author Response · Authors · 2021-07-29
> **Response to Reviewer Jwtr**
>
> Response to adding additional baselines:
>
> Thank you for this comment. This is a good point. We could have used embeddings from a PCNN, instead of R-BERT with BioBERT pre-trained weights, to form our relationship representations but Amin et al. (2020) show large improvements over Dai et al.'s (2019) work which makes use of the PCNN architecture. We believe an additional baseline using a PCNN would primarily illustrate the benefits of using embeddings from a large, pre-trained transformer over a PCNN, and it would only weakly support our case for abstractified multi-instance learning.
>
> Question 1:
>
> We experimented with sentence-level attention and observed similar results as Amin et al. (2020). This goes against our intuition and we hope to explore why an attention mechanism is not beneficial in this case and what can be done to make it more beneficial.
>
> Question 2:
>
> This is a good point -- for this task, sentence-level evaluation is not a good estimate of overall model performance. We only include sentence-level evaluation to highlight the benefits of AMIL over traditional multi-instance learning methods for predicting rare triples. For overall model performance, we use corpus-level evaluation.
>
> Question 3:
>
> This is true. The best and most fair comparison is between our proposed AMIL model and the previous Amin et al. (2020) model with SciSpacy since both models are trained and tested on identical data. The comparisons to Dai et al. (2019) and the original Amin et al. (2020) only help to illustrate the overall gains from our method using improved sentence segmentation, abstractified entities, and more informative relationship representations.

---

### Official Review · Reviewer_3SjD · 2021-07-21
**Valid contribution and encouraging results**

**Rating:** 6
**Confidence:** 3

**Review:**

The paper presents explores the benefits of abstractifying relationships, i.e., taking into account the type signatures of its instances, to improve relation extraction from biomedical text corpora. The paper applies this technique on a state-of-the-art method for relation extraction based on BioBERT. The experimental evaluation on a handful of relation representations suggests that abstractified multi-relational extraction with middle mention attention pools and entity end markers outperforms the state of the art, and is particularly useful for tail relationships, i.e., relationships for which there are not many sentences that assert them. The results show that for those triples, both the aforementioned techniques provide a performance boost (Table 3).

The paper is overall easy to read and the experimental results are encouraging. The experimental evaluation is also comprehensive. I have nevertheless a few minor remarks.

- Problem formulation

I think the paper lacks a clear and proper problem formulation, e.g., something like, given a text corpus T and a knowledge graph G, our goal is to extract a set X of triples, etc., etc. I managed to have a clearer picture of the problem rather late (when reading the experiments). Readers of the paper may benefit from an early formal problem formulation in order to know to which extent they could apply the techniques proposed in the paper.

- Datasets

While Section 3 suggests that the relationships "narrower" and "broader" were excluded for extraction, the paper enhances the underlying embedding model with abstractified triples, i.e., predicate signatures. Where do exactly those signatures come from? Is there an rdf:type/type predicate for entities?

Moreover, why did the authors exclude the relationships  "narrower", "broader", and "synonymous". Is it because they are trivial to extract? I think the paper should make clearer the rationale behind this decision.

Also, the paper should provide some hints of the quality of the corpus annotation process. How accurately are the entity mentions annotated? (Section 4.1). Studying this aspect might be interesting and useful for settings where structured knowledge is scarce and/or noisy.

- Evaluation

-- The paper states: "We also compare to Dai et al.(2019) but, due to a difference in data, comparisons to AUC and F1 are not possible but we are able to compare precision on the top K predictions."
I would recommend the authors to provide a more detailed justification.

-- The difference between corpus-level and sentence-level evaluation is not clear. Is this difference only based on the distinction between rare and common triples?

-- A website or git repository with access to the data and source code would be great!

---

> ### Author Response · Authors · 2021-07-29
> **Response to Reviewer 3SjD**
>
> Problem formulation:
>
> Thank you for the suggestion. We will add a problem formulation early in the manuscript so readers understand the problem without relying on other sections of the paper.
>
>
> Datasets:
>
> All the entity types and relationships come directly from the UMLS Semantic Network. In the Semantic Network, each entity has a unique entity type. Typically “broader”, “narrower”, and “synonymous”  are well-known static relationships between entities. For our task, we are interested in the more informative and potentially unknown relationships between biomedical entities to further scientific understanding. This decision is also informed by previous works on these tasks and allows for a more fair comparison between our model and past models. We will clarify this in the Dataset section of the manuscript.
>
> We appreciate the point made requesting a report on how accurately the entity mentions are annotated. We rely on the UMLS Metathesaurus, which contains concepts and their lexical variations, to identify entities from text. We believe it performs reasonably well and that a deeper analysis of its performance would require additional tests with a labeled dataset which is out of the scope of this paper. Analyzing the quality of the extracted concepts would indeed be informative and it is something we hope to do in the future.
>
>
> Evaluation:
>
> We were unable to attain the code and data used by Dai et al. (2019). Ideally, we would have trained and tested the Dai et al. model with the same data we used for our other experiments. Since this was not an option, we provide the results of the Dai et al. model as reported by the authors. Without access to data from their experiments, we believe a direct comparison is unfair. However, the precision at K indicates the model's overall ability to extract the true triples from a hold-out set of triples found in a test corpus. Because we use similar data (e.g. the UMLS knowledge graph with raw text from PubMed abstracts), we believe this metric allows for a good, but not perfect, comparison.  We will update the manuscript with the above explanation.
>
> We will work to clarify the difference between corpus-level and sentence-level evaluation. The key distinction is that corpus-level evaluation evaluates the model's ability to predict a hold-out set of unique triples from a test corpus. Corpus-level performance is used to gauge the model's usefulness in knowledge graph completion -- that is, predicting missing edges in a knowledge graph with high confidence. Sentence-level evaluation, or, in our case, bag-level evaluation, evaluates the model's performance on predicting the relationship for each bag of sentences. It's not typically used in biomedical relation extraction but it allows us to decompose triples by their frequency (e.g. the number of supporting sentences) into "rare" and "common" triples and report performance on each subset of triples. We include sentence-level evaluation to highlight the benefits of AMIL and confirm our hypothesis that grouping entities by entity type has a greater impact on the model's ability to predict "rare" triples since, with abstractified bags, we do not need to rely on upsampling as much as previous works.
>
> We plan to release all of our code on GitHub as well as the instructions for obtaining the data.

---

> > ### Comment · Reviewer_3SjD · 2021-07-29
> > **Thanks for the clarifications**
> >
> > I thank you very much the authors for the clarifications!

---

### Official Review · Reviewer_e1h2 · 2021-07-22
**Nice extension of multi-instance learning for biomedical RE supported by strong results**

**Rating:** 7
**Confidence:** 4

**Review:**

### Summary

This paper proposes a variant of multi-instance learning (MIL) for biomedical entity extraction called abstractified multi-instance learning (AMIL), which uses entity type information to construct bags of instances. In the paper, the authors first point out that a long-tail distribution of fact triples (i.e., (entity1, relation, entity2)) diminishes the advantage of the standard MIL since many bags might end up with duplicated instances (i.e., heavy upsampling is required to fill all the bags). This is particularly a problem in the biomedical data given that 50+% of fact triples appear in less than three sentences in PubMed. Instead, AMIL groups training instances by UMLS semantic types, which are shared across different entities. This approach results in more diverse instances in the bags, maximizing the benefit of MIL. For evaluation, the authors construct their own training/dev/test split from PubMed and UMLS. The BioBERT-base RE models are trained with AMIL and compare with the prior work (Amin et al., 2020) retrained on the same data. The main results show that AMIL models outperform the baseline by large margin. The authors further investigate performance on rare/common triples and 17 different relationship representations.


### Overall Review

Overall, this paper is clearly written and carefully explains the proposed methods and motivations behind them. The idea of AMIL is quite simple, but backing off to more general information is a clever solution to alleviate the data sparsity issues in biomedical RE data. Evaluations are fair, and the strong experimental results support the AMIL’s effectiveness. It’s interesting to see that diverse bags by AMIL (using type-level information) can gain over the standard MIL (using entity-level information) without losing performance. In addition, improvement on rare triples indicates that having diverse sentences in bags is crucial.  Although extensive architecture search on relationship representations is useful, it’s hard to see why “L” works best among others given the model’s black-box nature. The instance grouping is briefly mentioned in Section 4.1, but more detailed discussion would be helpful for reproducibility.


### Pros

1. Proposing AMIL that shows advantages on biomedical RE.
2. Experimental setup seems to be fair and appropriate. Analyzing performance on rare/common triples is informative.
3. Results are quite strong and promising.


### Cons

1. AMIL assumes an access to high-quality KBs such (e.g., UMLS) which might not be available in some data settings.
2. Although instance grouping is a key step of AMIL, it is not discussed in detail.


### Questions / Suggestions

1.	> "The UMLS semantic network is curated by human experts for decades and provides a rich ontology of biomedical concepts...”

      How would you evaluate the annotation quality of semantic types? What’s the average number of types per entity? Are they always consistent? Or, it depends on the editor? It would be nice to show some entities with types (possibly in appendix).

2. When you group training instances, do you always use all type combinations? Have you investigated good/bad type combinations? Do you end up with similar bags if many entity pairs share the same type sets?

3. Have you tried your version of MIL baseline (matching everything except grouping)?


### Minor points

- I see “Amin et al.” in many places (missing (2020)).
- In Table 2, the numbers from Dai et al. (2019) and Amin et al. (2020) are not directly comparable with your numbers (correct?). It would be nice to say it in the caption explicitly.

---

> ### Author Response · Authors · 2021-07-29
> **Response to Reviewer e1h2**
>
> Response to general comments:
>
> We agree, it is hard to see why relationship representation type "L" outperforms all other representations. Type "L" is a combination of the middle mention pool and entity end markers, both of which perform well on their own. In the paper, we lend intuition as to why the middle mention pool performs well but cannot do so with end entity markers. Determining why such an embedding performs best for biomedical relationship extraction would be an interesting area for future work.
>
> We will expand on the explanation of the instance grouping and we will release all of our code to ensure future efforts can easily recreate all of our experiments.
>
> The point that AMIL assumes access to high-quality KBs such as UMLS, which might not be available in some data settings, is a valid point and a key limitation to this approach and distant supervision in general. The method excels in the biomedical domain thanks to UMLS and could only generalize to other domains that have access to high-quality KBs. However, if approaches that rely on high-quality KBs, like AMIL, mature and become successful, a case can be made to divert human intelligence and resources to constructing KBs like UMLS rather than labeling text. We believe that the AKBC community will be interested in the idea of learning from a KB to construct a KB instead of learning from labeled text like what NLP community is attempting. Learning from a KB to construct a KB forms a positive feedback loop which we believe is a very promising direction for the field, in general.
>
> Question 1:
> Our understanding is that the UMLS semantic types are consistent and of reasonable quality. The Semantic Network is regularly maintained and updated by experts multiple times a year.  For entity types, we use the immediate parent of an entity. Each entity in the Semantic Network has a unique parent concept type. If accepted, we will add an example of an abstractified bag of entities to the Appendix.
>
> Question 2:
> This is an area we hope to explore in future work. We believe performance could be improved by identifying patterns of underperforming, or "bad," groups and potentially have multiple models to handle different grouping schemes. Further research is also needed to identify the level of similarity or dissimilarity between bags -- this will be an interesting area to look into.
>
> Question 3:
> Yes, we include the Amin et al. (2020) model with improved sentence segmentation (e.g. SciSpacy instead of the NLTK English) as a baseline biomedical relation extraction model that uses MIL without grouping. This baseline also uses the original relationship representation embeddings. As such, we believe it is a good baseline model to compare both AMIL and AMIL with improved relationship embeddings (e.g. Relationship Embedding Type "L").
>
> Response to Minor Points:
> We will address all these points in our revised manuscript. We will also add a formal problem formulation to the manuscript as Reviewer #2 also noted its absence.

---

### Decision · Program_Chairs · 2021-08-17

**Decision:**

Accept

**Comment:**

This paper proposes a variant of the canonical multi-instance learning (MIL) paradigm called abstractified MIL (AMIL), motivated in particular by the task of biomedical relation extraction. This paper presents a simple but clever approach to handling the "long tail" / data-sparse difficulties of distantly-supervised biomedical RE, which is clearly explicated and empirically evaluated, and which presents strong results. While reviewers had some questions around the dataset generation and evaluation protocols, and a few other minor aspects, the reception was overall very strong.